# Enhanced Recovery after Surgery (ERAS) in DIEP-Flap Breast Reconstructions—A Comparison of Two Reconstructive Centers with and without ERAS-Protocol

**DOI:** 10.3390/jpm12030347

**Published:** 2022-02-25

**Authors:** Sora Linder, Leonard Walle, Marios Loucas, Rafael Loucas, Onno Frerichs, Hisham Fansa

**Affiliations:** 1Department of Plastic Surgery and Breast Center, Spital Zollikerberg, 8125 Zollikerberg, Switzerland; sora.linder@spitalzollikerberg.ch (S.L.); marios.loucas@hotmail.com (M.L.); rafael.loucas@hotmail.com (R.L.); 2Department of Plastic, Reconstructive, and Aesthetic Surgery, Hand Surgery, Klinikum Bielefeld, OWL-University, 33604 Bielefeld, Germany; leonard.walle@klinikumbielefeld.de (L.W.); onno.frerichs@klinikumbielefeld.de (O.F.)

**Keywords:** autologous breast reconstruction, DIEP-flap, enhanced recovery after surgery (ERAS), length of stay, multicenter study

## Abstract

Enhanced recovery after surgery (ERAS) is established for autologous breast reconstruction. ERAS leads to a shortened hospital stay and improved outcome after elective surgery. In this retrospective, two-center case–control study, we compared two different treatment regimens for patients undergoing a DIEP-flap breast reconstruction from two centers, one with an established ERAS protocol and one without. All patients with DIEP breast reconstructions over the period of 12 months were included. The primary outcome measure was the length of hospital stay (LOS) in days. A total of 79 patients with 95 DIEP-flaps were analyzed. In group A (ERAS) 42 patients were operated with DIEP flaps, in group B (non-ERAS) 37 patients. LOS was significantly reduced in the ERAS group (4.51 days) compared to the non-ERAS group (6.32; *p* < 0.001). Multivariate analysis showed that, in group A, LOS is significantly affected by surgery duration. BMI in the ERAS group had no effect on LOS. In group B a higher BMI resulted in a significantly higher LOS. In multivariate analysis, neither age nor type for surgery (primary/secondary/after neoadjuvant therapy, etc.) affected LOS. In both groups, no systemic or flap-related complications were observed. Comparing two reconstructive centers with and without implemented ERAS, ERAS led to a significantly decreased LOS for all patients. ERAS implementation does not result in an increased complication rate or flap loss. Postoperative pain can be well managed with basic analgesia using NSAID when intraoperative blocks are applied. The reduced use of opioids was well tolerated. With implementation of ERAS the recovery experience can be enhanced making autologous breast reconstructions more available and attractive for various patients.

## 1. Introduction

Enhanced recovery after surgery (ERAS) guidelines are now established for many surgical fields. The perioperative measures are also described in breast cancer surgery and autologous breast reconstruction [1,2,3,4,5,6,7,8]. According to Kehlet et al. ERAS leads to a shortened hospital stay and improved outcome after elective surgery [9].

Shortened length of stay (LOS) reduces the risk of nosocomial infection, places less psychological burden on patients, and also reduces costs.

However, routine autologous microsurgical breast reconstruction is still a complex procedure, and several components contribute to its complexity and outcome: cancer diagnosis with possible neo-adjuvant chemotherapy, two to four surgical sites, microsurgery, anesthesia longer than 3 h, pain management and delayed mobilization. The mentioned factors possibly decrease patients’ acceptance and referral. Therefore, many patients who would benefit from autologous reconstruction are not primarily included.

To enhance acceptability and outcome at one center (A), an enhanced recovery protocol for microsurgical breast reconstruction was implemented in 2019. Preoperative information about the procedure was comprehensive, intraoperative measures were taken to reduce opioid consumption, allow early postoperative mobilization, and postoperative pain management was without opioids. Center B did not implement any special ERAS measures.

Therefore, the purpose of this study was to report the clinical outcomes, complications, and reoperations in a series of patients undergoing a DIEP-flap breast reconstruction and evaluate whether an ERAS protocol can contribute to a reduced LOS.

## 2. Patients and Methods

In this retrospective, two-center case–control study we compared two different treatment regimens for patients undergoing a DIEP-flap breast reconstruction from 2 centers (one in Switzerland, one in Germany), one with an established ERAS protocol and one without. Both centers are well experienced in autologous breast reconstruction. In each group surgery was only performed by the same senior surgeon team.

The hospital records of patients who underwent a deep inferior epigastric perforator flap (DIEP) based breast reconstruction were examined. All patients who underwent a primary or secondary, unilateral or bilateral DIEP breast reconstruction from September 2020 until September 2021 were included. The patients were unselected and consecutive.

The primary outcome measure was the length of hospital stay (LOS) in days. Patient characteristics, age and BMI were collected; additionally, breast cancer surgery data were collected to distinguish between uni- or bilateral surgery, primary or secondary reconstructive procedures, surgery after neoadjuvant treatment and simultaneous lymphatic surgery.

The ERAS standards that were implemented in group A hospital contain an intensive preoperative preparation of the patient in terms of information and consent. The patient was informed on the enhanced postoperative protocol and a planned discharge after 3 to 4 days. CT scans were evaluated to map the DIEP-perforators and the course of the epigastric artery and vein. The ERAS standards involved furthermore improved intraoperative surgical procedure, perioperatively the anesthesia was maintained with propofol, titrated according to bispectral index (BIS). Sufentanil or remifentanil were administered as bolus on demand, avoided if possible. The neuromuscular blockade was obtained with rocuronium. Vasopressors were avoided, if necessary, noradrenaline was chosen. I.v. fluid supply was limited to 100 mL/h. Ropivacaine (5 mg/mL) was generously administered to the surgical sites and continuous pain treatment was started during surgery with i.v. paracetamol and continued orally after operation. Postoperatively the patients were monitored in the post-anesthesia unit and transferred to the regular surgical ward, where they received scheduled administration of paracetamol and ibuprofen.

While performing the surgery, gentle tissue handling was of the highest importance. A defined 2-team approach was chosen to decrease operation time. The incision of the rectus sheath was limited to 4–5 cm (Figure 1). If available, a perforator vessel was chosen as recipient vessel, so that intercostal dissection and removal of ribs could be avoided with further reduction in operation time and postoperative pain. If no perforator was available, a rib sparing approach was chosen when possible. The artery was hand-sewn with Ethilon 9-0 BV4 needle (Ethicon, Norderstedt, Germany) and the vein was always couplered (Synovis, Birmingham, AL, USA). The intraoperative administration of ropivacaine at the harvest and recipient site was performed under visual control [10]. Abdominal quilting sutures allowed surgeons to abandon drainage; drains were only placed in the breast. Skin islands on the flap were only placed when required for the breast shape. If the flap was intraoperatively well perfused, a monitor island was omitted and flap-monitoring was not performed. All patients were mobilized the same evening after surgery, latest on the first postoperative day. The foley catheter was directly removed after surgery, latest on the evening of the operative day. I.v. lines were removed after the patient tolerated her first meal without nausea. All patients received nutritional counseling and a high protein supplement (2× day). The drains were removed as soon as possible, latest at hospital discharge.

The patients without ERAS protocol in center B were treated according to the traditional regime. Anesthesia was maintained with sevoflurane and sufentanil. The neuromuscular blockade was obtained with cisatracurium besylate. Vasopressors were avoided, if necessary, noradrenaline was chosen. There was no i.v. fluid restriction.

Initial postoperative analgesia was achieved with paracetamol 1 g i.v. in a 15 min intravenous infusion, and additionally with piritramide, an opioid analgesic, according to pain levels. The patients were transferred back to the surgical ward; the pain treatment consisted of ibuprofen orally and additional paracetamol and hydromorphone as a reserve drug. The mobilization of the patients started 1 day postoperatively after an initial bed rest. The foley catheter was removed 1 or two days after the operation.

Routine blood counts and CRP were obtained on days 2 and 7. The patients were followed up after discharge at the office at 1, 2 and 6 weeks and then after 6 months or if further surgery was planned.

### Statistical Analysis

The two cohorts (group A and group B) were compared using an independent-samples *t*-test for continuous variables and chi-square analyses for categorical variables to detect differences between the groups. The normal distribution of variables was tested with the Shapiro–Wilk test and compared data with the paired t-test (parametric data) or the Wilcoxon ranksum test (non-parametric distribution). Fisher exact test was used for categorical variables. A difference of *p* < 0.05 was considered to be statistically significant. Multivariate analyses (Spearman and Pearson/Point biserial analysis) were used to identify interacting factors and correlations. All the statistical analyses were conducted using the SPSS software v28.0 (IBM, New York, NY, USA).

## 3. Results

### 3.1. Patients and Demographics

A total of 79 patients with 95 DIEP-flaps were analyzed in the study. In group A (ERAS), 42 patients were operated with DIEP flaps, and in group B (non-ERAS) 37 patients. In group A, a total of 48 flaps were operated, of which 36 were unilateral and 6 bilateral. In group B, 47 flaps were operated, of which 27 were unilateral and 10 bilateral. A total of 14 flaps in group A had skin islands due to oncological (e.g., replacement of nipple-areolar complex) or aesthetic reasons (e.g., skin needed for breast shape), all other flaps had no monitor island. In group B, all flaps had monitor islands and were monitored postoperatively. All flaps had one arterial and one venous anastomosis, except two where one additional venous anastomosis (also with a coupler) was performed. This had no influence on LOS. In both groups, no systemic or flap-related complications were seen clinically, no revisions or readmissions were indicated. In all patients, the lab work showed no pathological findings. There was also no late revision. Figure 2 and Figure 3 show examples from the ERAS group.

In terms of demographics and comorbidities, the two groups did not differ significantly. In both groups, approximately 2/3 of the patients were primary reconstructions. Detailed demographic data are presented in Table 1.

### 3.2. Clinical Outcome

ERAS protocol in patients undergoing a DIEP-flap breast reconstruction (group A) resulted in a considerable reduction in LOS. This was significant when comparing the ERAS group (mean 4.51, SD 1.4, days) and the non-ERAS group (mean 6.3, SD 1.3; *p* < 0.001, Figure 4).

This was also significant when comparing the groups regarding unilateral reconstruction (4.4 vs. 6.3). The difference became not significant between the groups for the patients with bilateral reconstructions (5.3 vs. 6.5). The same observation is seen for the differences in operation time (minutes): in unilateral reconstructions, a significant difference between the ERAS group (mean 224.5, SD 52.6) and the non-ERAS group (mean 335.7, SD 63.2) was found (*p* < 0.001), while the surgical time difference was not significant in bilateral reconstructions (Table 2). The multivariate analysis showed that, in group A, LOS is significantly affected by surgery duration. This dependence does not exist in group B without ERAS.

BMI in the ERAS group had no effect on LOS. In group B, however, a higher BMI resulted in a significantly higher LOS. In group A, LOS is significantly affected by surgery duration. This dependence does not exist in group B without ERAS. Incidentally, the analysis revealed that in both group surgical duration is significantly influenced by BMI. The higher the BMI, the longer the operation.

In multivariate analysis, whether surgery was primary or secondary, after neoadjuvant therapy, or included lymphatic surgery did not affect LOS. Additionally, age did not show an influence on LOS.

## 4. Discussion

ERAS includes a variety of peri-operative measures. Not every step is applicable to every patient. However, the sum of all possible measures leads to an improvement in patient care and reduce complexity.

In both groups, the respective operations were successful. There was no flap loss and no readmission of the patients. The differences between the two groups are therefore in the management of perioperative measures and intraoperative techniques. With the structured implementation of ERAS, the time spent in the hospital can be shortened, thus relieving the patient psychologically and improving outcome.

ERAS led to a significantly decreased LOS for all patients, especially in unilateral reconstructions. Considering the group of bilateral reconstructions alone, ERAS shows no significant advantage related to hospital stay. This shows the limitations of ERAS-related improvements in this surgery. However, for the most common standard case of unilateral reconstruction, ERAS results in a significant reduction in LOS.

The multivariate analysis shows that in group B, BMI significantly increased LOS. The higher the BMI, the longer the hospital stay. Almost all studies show increased LOS, complications and costs with increasing BMI. In the ERAS group, there was no significant effect of BMI on LOS [11]. This effect was also shown by other study groups [12]. Thus, ERAS can remove a major risk factor for complications in autologous breast reconstructions. Here, we showed that in the ERAS group, LOS was also influenced by the duration of surgery, suggesting that essential measures of ERAS are not only perioperative but also intraoperative surgical.

An important element is the psychological support of the patient in relation to the surgery. Many patients are frightened by the primary diagnosis and the surgery, which involves a mastectomy on the one hand and a lengthy medical procedure on the other. They are worried that they will lose their femininity, but at the same time read about the procedure on the internet that it takes a long time and may not always go well. Establishing ERAS is therefore a way to present to patients that breast reconstruction is a routine procedure. This includes the preoperative information and the standardized, relatively short inpatient length of stay, which relieves patient’s anxiety. Here, the patient’s expectation is positively influenced so that they also perceive the procedure as a “simple” procedure [13].

**Flap monitoring**. Traditionally, flaps are monitored postoperatively. This dates from a time when microsurgical procedures were not routine. Other ERAS protocols continue to monitor hourly for the first 2 days [14]. However, reconstructive breast surgery is now highly standardized, and most operations are performed in centers with appropriate surgical experience. Therefore, in our opinion, it is legitimate not to perform postoperative flap monitoring. As can be seen from the two study groups, the insertion of a monitor island does not have a positive effect, nor does the omission of a monitor island that is not necessary lead to a problem. As shown in previous publications from our group, when flap monitoring was part of the concept in our general patient population, the success rate is over 98% [15]. The experience of the last decades has shown that flaps for breast reconstruction that are vital on the operating table after anastomosis continue to stay vital [16]. Prerequisites are (1) successful venous and arterial anastomoses, (2) good position of the pedicle (no kinking, no traction) and good fixation of the flap allowing mobilization of the patient, (3) good initial flow (measured by blood pressure) by catecholamines (e.g., norepinephrine) and (4) avoidance of vasospasm. Additionally, patients’ history should not note any thrombosis indicating thrombophilia [16]. Therefore, in the ERAS group, monitor islands were not used unless flap skin was required as an integrative part of breast reconstruction. This facilitates postoperative handling, especially in nipple-sparing mastectomies. Patients return to the regular ward postoperatively, and flap monitoring occurs only for flaps that are unsafe on the operating table. Postoperatively, only pain, swelling, and drainage fluid quantity are recorded by nursing staff. Monitoring via Doppler is also not performed. As a result, most patients spend an undisturbed first night, do not feel insecure and anxious, and are better mobilized, which also reduces the risk of thromboembolism.

An essential pillar of an ERAS protocol is early, pain-reduced mobilization. An attempt should always be made to mobilize the patient in the ward on the day of surgery. This assumes that the patient is circulatory stable, has no nausea, and does not experience severe pain during mobilization. Mobilization is more difficult if the patient is receiving strong pain medications that impair circulation or cause nausea [5]. Therefore, to positively influence the postoperative pain situation, intraoperative application of long-lasting local anesthetics is a good way to reduce pain [17,18,19]. Usually, these allow early mobilization without the need to give opioids.

Additionally, the least possible number of drains and infusions facilitate mobilization. In the used ERAS protocol, intraoperative administration of ropivacaine has proven effective. This is injected into the rectus and externus sheath and the abdominal skin, and at the same time we do not use drains on the abdomen, without having seen any negative effect (increased seromas or infections) in our patient population. Ropivacaine is also infiltrated in the thoracic region (cranial und lateral pectoral muscles, serratus muscle, axilla) and also at the outlet of the drainage, which is almost always required. As a long-acting local anesthetic, ropivacaine has been shown to work better than bupivacaine in some studies, although the latter has become also a valuable component of many ERAS protocols in its liposomal version [17,20].

In addition, in group A, the foley catheter is removed at the end of the operation. This reduces the risk of urinary tract infection and, at the same time, increases the patients’ early mobilization by going to the bathroom. Early mobilization, in turn, reduces the risk of thromboembolism and reduces constipation which impairs quality of life and increases abdominal pain after flap harvesting.

**Surgical technique**. As mentioned above, the surgical technique has not been considered so far in ERAS protocols. While this is more standardized in visceral surgery, such details differ greatly in autologous breast reconstruction. Although the microvascular connection to the recipient site is relatively standardized at the internal mammary artery and its accompanying veins, there are gradations here as well. In our two groups, ischemia time was not significantly different, indicating a standardized anastomosis process. Whenever possible, in group A (ERAS) the anastomoses were performed to the perforator vessels of the mammary artery and vein, or if this is not possible, a rib-sparing approach to the IMA/V was chosen. These approaches are always less painful than those that involve removing the rib over the vessels [15].

At the abdomen also different techniques play a role regarding the pain. For example, Lockwood’s limited immobilization of the abdominal wall is likely to cause less pain than wide mobilization laterally beyond the costal arches. In addition, in the ERAS-group, surgical technique kept the incision to the rectus sheath to harvesting a DIEP-flap as short as possible, trying to get by with an average length of 4–5 cm fascial incision [16]. Elevation of an ms-TRAM usually requires a larger muscle and fascial incision, so pain may increase here. The insertion of a mesh to stabilize the abdominal wall may also lead to increased postoperative pain and increased hospital stay. Therefore, with preoperative planning by CT, the ERAS group focused on reducing pain by using a single perforator DIEP flap and avoiding a mesh. Additionally, in the ERAS group, no drains on the abdomen were used.

With the introduction of the diagnosis-related groups (DRG) system in hospital billing in Switzerland and Germany, the LOS in hospital has also been reduced, because the earning is flat per case but not per actual cost. For Switzerland, the DRG for a unilateral DIEP-flap offers a relative weight of 2.8, the hospital earns approximately CHF 27.000 (currently: EUR 25.800) per case. In Germany, the same operation is rated with a relative weight of 3.4, earning approximately EUR 12.770. ERAS does not incur any additional costs; only additional costs for nutritional counseling and targeted physiotherapy may occur. Thus, the hospital benefits from the lower costs of the shorter LOS after ERAS has been established.

The reduced LOS trend was also seen from the American College of Surgeons National Database, where LOS after unilateral reconstructions decreased from 4.47 days in 2012 to 3.9 days in 2018 [21]. In a single center cohort study from the Netherlands, ERAS reduced the LOS from 6.2 to 5 days [14]. Here, we have shown a LOS of 4.51 days for the ERAS and 6.32 days for the non-ERAS groups. Disparities between LOS in the U.S. and European countries may be explained by differences in medical insurance and societal attitudes. From our point of view, the patient should always be able to care for herself after discharge. Thus, most patients will spend 3–4 days in the hospital. Secondary reconstructions and younger patients are the ones with shorter stay.

One limitation of this study is the retrospective design; nevertheless, the follow-up and data assessment of the patients occurred on a regular and prospective basis with a standardized follow-up postoperatively as both groups were comparable and no changes in staff or (post)-operative workflow had been performed. A prospective study could have evaluated more parameters involved. Additionally, the sample size regarding additional parameters such as neo-adjuvant treatment or lymphatic surgery can be increased with a longer observation time span. However, prospective studies are costlier and more labor-intensive in regular clinical settings.

## 5. Conclusions

Comparing two reconstructive centers with and without implemented ERAS protocol, ERAS led to a significantly decreased LOS for all patients, especially in unilateral reconstructions. ERAS implementation does not result in an increased complication rate or flap loss. Revision rates and readmission rates are identically between groups. Postoperative pain can be well managed with basic analgesia using NSAID when intraoperative blocks are applied. The reduced use of opioids was well tolerated. With implementation of ERAS the recovery experience can be enhanced making autologous breast reconstructions more available and attractive for various patients.

## Figures and Tables

**Figure 1 jpm-12-00347-f001:**
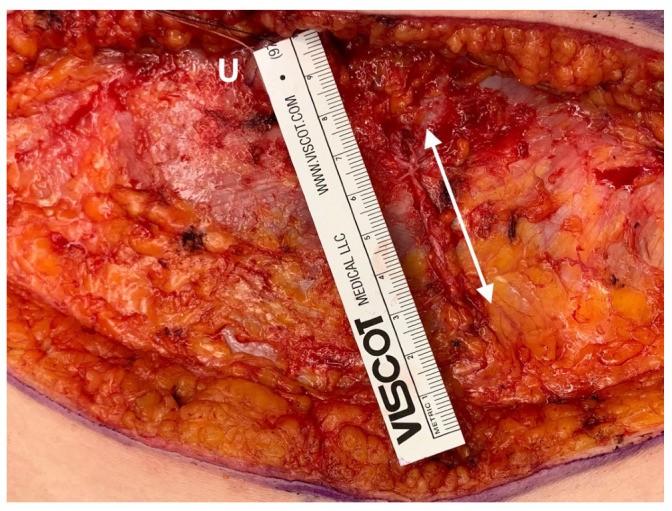
Intraoperative view of DIEP-flap donor site (ERAS-group). Rectus abdominis sheath incision sutured, marked with white double arrow, approximately 5 cm length. Umbilicus (marked with U) hidden behind the upper part of ruler.

**Figure 2 jpm-12-00347-f002:**
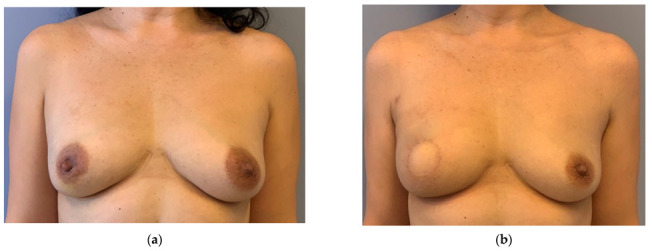
(**a**) Preoperative: 46 years old, otherwise healthy patient with invasive breast cancer on the right side. She underwent a skin-sparing mastectomy, sentinel-lymphonodectomy and immediate reconstruction with DIEP-flap (ERAS-group). (**b**) Postoperative: Patient seen 6 months after surgery and radiation therapy. The skin island replaces the areola. Operation time 157 min, ischemia time 42 min, 2.5 mm coupler, flap weight 420 g, length of stay 4 days.

**Figure 3 jpm-12-00347-f003:**
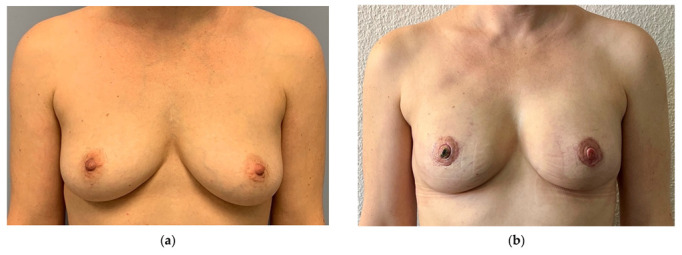
(**a**) Preoperative: 43 years old, otherwise healthy patient received a bilateral prophylactic nipple-sparing mastectomy and immediate reconstruction with DIEP-flap (ERAS-group). (**b**) Postoperative: Patient 6 weeks after surgery. No monitor island was placed, small nipple necrosis on the right side was treated conservatively. Operation time 348 min; ischemia time 37 min right, 44 min left; 2.0 mm coupler right, 2.5 mm left; flap weight 290 g right, 285 g left; length of stay 5 days.

**Figure 4 jpm-12-00347-f004:**
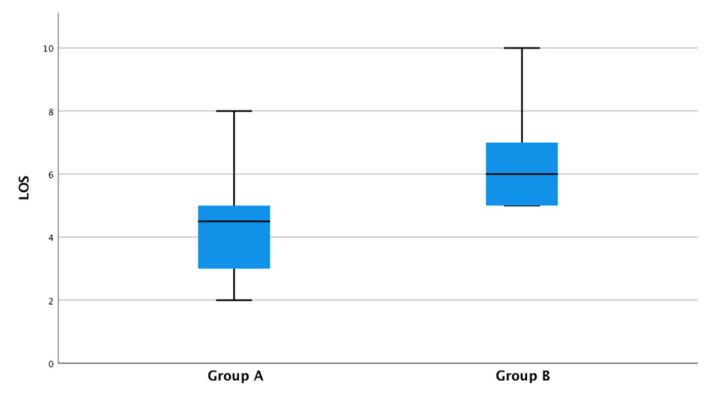
In group A with ERAS (mean 4.51, range 2–8, SD 1.4) compared with the group B without the ERAS (mean 6.23, range 5–10, SD 1.27).

**Table 1 jpm-12-00347-t001:** Demographic data for the study groups.

	DIEP Group A (ERAS) Mean ± Std (Min, Max)	DIEP Group B (Non-ERAS) Mean ± Std (Min, Max)	*p*-Value	Test
Age at surgery	52.66 ± 11.54 (32; 78), *n* = 42	38.33 ± 9.86 (27; 78), *n* = 37	0.462	MWU
Gender	42 Female	37 Female	-	Fisher
Side (Uni/Bilateral)	36 Unilateral/6 Bilateral	27 Unilateral/10 Bilateral	-	Fisher
DIEP-flaps	48	47	-	Fisher
Body Mass Index	25.83 ± 4.48 (18.2; 35.6), *n* = 42	25.02 ± 3.74 (17.09; 35.1), *n* = 37	0.194	MWU
Primary/Secondary DIEP-flaps	28/14	25/12		

**Table 2 jpm-12-00347-t002:** Intra- and postoperative results in group A (ERAS) and B (non-ERAS) including length of stay, surgical time, ischemia time, coupler size, revisions of the 79 patients available for personal follow-up. Values are given as mean ± SD [95% confidence interval] or (minimum; maximum).

	DIEP Group A (ERAS)	DIEP Group B (Non-ERAS)	*p*-Value
Length of stay (LOS, days)	4.51 ± 1.41 (2; 8)	6.32 ± 1.27 (5; 10)	<0.001
Length of stay (LOS, days) Unilateral	4.38 ± 1.07 (3; 7)	6.25 ± 1.05 (5; 9)	<0.001
Length of stay (LOS, days) Bilateral	5.33 ± 2.73 (2; 9)	6.5 ± 1.77 (5; 10)	0.158
Surgical time (min) Unilateral	224.52 ± 52.59 (157; 376)	335.71 ± 63.24 (242; 458)	<0.001
Surgical time (min) Bilateral	381.5 ± 19.75 (348; 404)	403.64 ± 76.39 (380; 647)	0.019
Ischemia time (min)	43.37 ± 7.23 (34; 59)	41.70 ± 5.27 (31; 60)	0.224
Revisions	*n* = 0	*n* = 0	-
Readmissions	*n* = 0	*n* = 0	-

## Data Availability

Supporting data is available from the authors upon request.

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
