# Peer review of "Enhanced Recovery after Surgery (ERAS) in DIEP-Flap Breast Reconstructions—A Comparison of Two Reconstructive Centers with and without ERAS-Protocol"

_jpm, 2022, doi:10.3390/jpm12030347_

Round 1

Reviewer 1 Report

The authors review Enhanced recovery after surgery (ERAS)for autologous breast reconstruction and its benefits, primarily on length of stay. The methodlogy adequately investigates two patient groups with or without ERAS. The surgical results are excellent and the paper is very informative.

Small changes need to be made.

Introduction:

Some part of the example file has wrongly been incorporated:

“The introduction should briefly place the study in a broad context and highlight why … line 54 and so forth

Methods:

Please name how many vessels are sutured and if coupler devices are used.

How long was the observation performed and how regarding flap loss, if no routine monitoring is performed?

Please state the follow up regarding readmission and flap loss.

How do you discover the 2% flap loss described in the discussion without monitoring?

Results

how many flaps were buried? How many had flap skin for possible monitoring

Overall the manuscript is very well written and the work very informative.

Author Response

Thank you very much for your very kind and constructive review. All changes are marked in red within the manuscript. 

  1. We are sorry for not deleting the parts from the template. This was corrected.
  2. In both groups we have always hand-sewn the artery, and coupled the veins. This was stated in the text.
  3. We included how long the patients were followed. Besides clinical controls (such as swelling, inflammation etc) we controlled leucocytes and CRP as indicators for arising problems. Additionally, the patients were seen in the office, one, two and six weeks after surgery. This was also stated in the text including readmissions and flap loss.
  4. From our previous analysis (more than 500 cases) we have seen a microvascular success rate of more than 98%, less than 2% loss. This is constant with the literature. With that previous series we have always used monitor islands before changing to use skin island only when aesthetically or oncological needed. We state that in the text.
  5. We also added how many flaps were buried and how many had a skin island that could be monitored. 

We hope that the paper now meets the criteria of the reviewer. For your information the second reviewer suggested to include pictures of clinical outcomes which was also done. 

Reviewer 2 Report

Thank you for the opportunity to review this manuscript. This an interesting study. The authors attempt to compare two reconstructive centers with and without implemented ERAS, and they conclude that ERAS led to a significantly decreased LOS  for all patients. This reviewer believes the following comments need to be addressed:

1- Please comment on the cost of each approach

2- Please include clinical pictures and clinical outcomes

Thanks again for the opportunity to review this manuscript

Author Response

Thank you very much for your kind and constructive review.

  1. We have expanded the part on the cost in the "Discussion". It was marked in red in the text. 
    "With the introduction of the diagnosis related groups (DRG) system in hospital billing in Switzerland and Germany, the LOS in hospital has also been reduced because the earning is flat per case but not per actual cost. For Switzerland the DRG for a unilateral DIEP-flap offers a relative weight of 2.8, the hospital earns approx. 27.000 CHF (currently: 25.800 €) per case. In Germany the same operation is rated with a relative weight of 3.4, earning approx. 12.770 €. ERAS does not incur any additional costs, only additional costs for nutritional counseling and targeted physiotherapy may occur. Thus, the hospital benefits from the lower costs of the shorter LOS after ERAS has been established
  2. We have now included clinical pictures of the outcomes. One picture of the donor site area, one unilateral and one bilateral DIEP-flap reconstruction.

We hope the paper now meets the criteria of the reviewer.